# Situational Awareness in the Context of Clinical Practice

**DOI:** 10.3390/healthcare11233098

**Published:** 2023-12-04

**Authors:** Shani Feller, Liviu Feller, Ahmed Bhayat, Gal Feller, Razia Abdool Gafaar Khammissa, Zunaid Ismail Vally

**Affiliations:** 1University of Witwatersrand, Fir Avenue, Bantry Bay, Cape Town 2050, South Africa; 2Department of Community Dentistry, School of Dentistry, University of Pretoria, Pretoria 0084, South Africa; 3Department of Radiation Oncology, University of Witwatersrand, Johannesburg and Charlotte Maxeke Academic Hospital, Johannesburg 2193, South Africa; 4Department of Periodontics and Oral Medicine, School of Dentistry, Faculty of Health Sciences, University of Pretoria, Pretoria 0084, South Africa; 5Department of Prosthodontics, School of Dentistry, Faculty of Health Sciences, University of Pretoria, Pretoria 0084, South Africa

**Keywords:** situational awareness, clinical reasoning, clinical judgement, decision making, mental models, electronic health records, pattern recognition, information processing

## Abstract

In the context of clinical practice, situational awareness refers to conscious awareness (knowledge), which is a mental model of a given clinical situation in terms of its elements and the significance of their interrelation. Situational awareness (SA) facilitates clinical reasoning, diagnostic accuracy, and appropriate goal-directed performance, and it enables clinicians to immediately adapt treatment strategies in response to changes in clinical situational actualities and to modify the course of goal-directed activities accordingly. It also helps clinicians prepare future operational plans and procedures based on the projection of situational developments. SA, therefore, is an important prerequisite for safe clinical procedures. The purpose of this narrative review is to highlight certain cognitive and external (environmental) situational factors that influence the development of situational awareness. Understanding the dynamic, adaptive, and complex interactions between these factors may assist clinicians and managers of healthcare systems in developing methods aimed at facilitating the acquisition of accurate clinical situational awareness and, in turn, may bring about a reduction in the incidence of SA, diagnostic, and operational errors.

## 1. Introduction

Situational awareness (SA), domain-specific clinical knowledge and technical skills, and non-technical skills, such as teamwork, communication, cognition, and self-control, are some of the most important factors contributing to the efficient and effective processes of clinical judgment, decision making, and subsequently, synchronized goal-directed activities [1,2]. SA is a dynamic state of conscious awareness [2,3] that is brought about by complex interactions between multiple cognitive factors, including attention, working memory, long-term memory, cognitive flexibility, and rule acquisition [2,4] and by drawing from pre-determined mental models that have been developed via repeated relevant situational experiences and clinical training [4].

SA can be viewed as one’s knowledge about a given situation's elements and circumstances [5,6,7], and this conscious awareness plays an important role in a decision maker’s ability to manage complex and dynamic external tasks through adaptive goal-directed actions [3]. It facilitates the process of decision making in the face of rapidly changing situational actualities and enables decision makers to anticipate and subsequently manage unplanned task descriptions and goal-directed demands under uncertain operational circumstances [4].

A complex clinical situation generates multiple internal representations, some of which may be contradictory, and each such internal representation may require a different goal-directed action to solve the clinical problem, thus contributing to the uncertain nature of clinical decision making [8]. The integration of successive new pieces of relevant situational information and the identification and resolution of conflicting perceptions are essential for developing and maintaining high-level SA and enable clinicians to form and support an accurate, comprehensive mental model of a situation [6,9,10].

In the context of clinical practice, SA—that is, the integrated representation of a clinical problem—is essential for the processes of clinical reasoning, judgment, decision making, and performance [6,11,12], and it enables the anticipation of evolving changes in current situational actualities and task descriptions based on similarities with past clinical situations. It also enables the projection of the risk–benefit trade-offs of the current operational activity to future clinical situations [5].

Many factors may contribute to the faulty detection, capturing, and processing of situational clinical data and information, which may lead to the construction of an inaccurate SA with subsequent diagnostic errors and unfavorable treatment outcomes. These factors include mental exhaustion resulting from excessive cognitive demands associated with data and information processing and from a decrease in resources of mental energy; exposure to work overload and to work-related stressors, task saturation, fatigue, burnout, and negative emotional stimuli; and cognitive activities that divert attention away from goal-directed tasks [6,7,9,13,14]. On the other hand, the accuracy of SA may be enhanced by using electronic health records, clinical decision support systems, and telemedicine, all of which reduce cognitive loads, ease clinical judgment and decision making, and support SA [4,15].

A clinician’s SA of a given clinical problem is based on information obtained from the patient’s history, physical examinations, test results, images, and consultations with a variety of healthcare professionals, and it is influenced by the clinician’s cognitive capacities in relation to information processing and clinical reasoning, which rely on intuitive, non-analytical, and deliberate analytical reasoning pathways [16]. Data that are unknown, uncertain, or ambiguous, the failure of communication systems and of information processing systems, inadequate knowledge, weaknesses in relevant mental models, and overconfidence are some additional clinician-related factors that may impair the development of accurate SA [17].

The process of acquiring accurate SA is highly susceptible to attentional impairment, cognitive fixation, and cognitive biases, but post-action reflective and metacognitive practices may improve the capacity to develop and maintain SA [1]. Cognitive distractions and interruptions induced by extrinsic or intrinsic stimuli can reduce or shift focused attention away from the target objectives and diminish the ability to hold information in the working memory. These factors, together with the fixation on one internal representation to the exclusion of others, negatively impact SA [5].

Maintaining focus and conscious awareness of a given clinical situation is essential for detecting diagnostic errors and for adapting the initial operative plans according to the dynamic course of the clinical situation [18]. Clinicians in the fields of anesthesiology, emergency medicine, surgery, and intensive care deal with highly dynamic, complex, and risky clinical cases; therefore, they should have high levels of SA to enable them to maintain safe and effective goal-directed activities [5]. The process of acquiring accurate SA can be facilitated by cognitive engineering systems that are able to construct and present relevant clinical data and information in a format conducive to the development of mental models (a mental model is an internal representation of and knowledge about an external situation, and it is used to plan and execute goal-directed tasks) [19].

Discussions about how to acquire SA and the practical training therein should be incorporated into the curriculum of under- and post-graduate students of clinical health sciences and into the quality-assurance policies of healthcare providers with a view toward increasing awareness of this phenomenon and subsequently reducing the frequency of SA errors. The purpose of this narrative review is to contribute to a better understanding of the role that SA plays in the context of clinical activities, including diagnosis, planning, decision making, and the execution of goal-directed clinical operations. The information for this narrative review was obtained by employing the PubMed and MEDLINE database search engines with the search terms: situational awareness, information processing, pattern recognition, mental models, clinical reasoning, clinical judgment, clinical decision making, medical uncertainties, analytical reasoning, intuitive reasoning, and health information technology; in addition, references from relevant English language articles that were deemed pertinent were analyzed.

## 2. The Situational Awareness Construct

The framework of situational awareness (SA) refers to SA as an evolving state of knowledge about the dynamically interacting elements of a given situation [7,9]. According to this framework, all situational knowledge is generated by sequential cognitive steps that promote information processing, and it enables predictions of the immediate future state of an evolving situation (Figure 1) [7,20]. SA can be viewed as a comprehensive and coherent internal representation of any current situational actualities; it is continuously assessed and updated in accordance with the dynamic state of the situation [10,21] and is imprinted in the consciousness of the decision maker (in our case, the clinician) [3,20].

Thus, SA should be regarded only as a description or label of an internal representation of a given situation and not as a cognitive mechanism or process in itself [9,21,23]. The cognitive mechanisms that play active roles in developing a meaningful conscious awareness about a given situation include long-term memory, short-term memory, attention, and other executive functions, but not SA itself [7]. Situational awareness is not an empirical reality but an abstract construct; a failure of a clinical operation associated with deficient SA cannot be attributed to an ‘impaired’ SA but rather to the imperfect functional activity of the psychological mechanism that generated it [24].

In the context of clinical practice, the specifications of what is considered to be an adequate state of knowledge that constitutes a fully developed SA are not well defined [20]; therefore, it is difficult to measure and assess this clinical phenomenon [25]. Further, in order to have operational meaning, SA has to facilitate the understanding and the meaning of the interactions among the clinical situational elements (signs, symptoms, etc.) so that clinical judgments, decision making, and goal-orientated activities are efficient and effective [21].

Situational factors that may negatively impact the accuracy of a given SA and thus may facilitate the generation of SA errors include uncertain, complex, incomplete, inaccurate, or misrepresented relevant clinical information; clinician-specific factors include limited professional knowledge and clinical expertise, uncertainty intolerance, and imperfect mental acuity (i.e., memory, attention, and executive functions) [26,27], all of which hinder information processing, clinical judgments, decision making, and the projection of future situation-related dynamics, and may lead to compromised execution of clinical tasks.

The behavioral response to situational actualities is determined by the constraints of a situation (stimuli) and how these affect the observer’s perception and action, as well as by the intrinsic constraints of the observer. Consequently, the phenomenon of SA is determined by the properties of a given situation and by the experience (awareness) of the observer evoked by the stimuli [23]. The experiential awareness component of SA is brought about by information-processing cognitive elements, including perception, comprehension, and projection (Figure 1), that enable clinical judgement, decision making, action, and anticipation of future situational eventualities in the context of clinical practice [2,23,26].

In general, the cognitive pathway by which information is processed in order to acquire high-level SA is related to the properties of the situation (complexity, dynamics, typicality, uncertainty) and to the clinician’s cognitive acuity, experience, and expertise. The more complicated a situation is, the more difficult it is to develop and maintain high-level SA; the less experienced the clinician is, the greater the need for deliberate and analytical knowledge-based processing. However, in general, in order to acquire and maintain high-level SA, the clinician usually has to use a combination of the different information-processing pathways (see below).

## 3. Information-Processing Procedures in Relation to Situational Awareness

Reasoning is a cognitive activity that uses information-processing procedures to drive rational processes in order to solve problems; the dynamic and interconnected relationships between data, information, and knowledge are a fundamental prerequisite for clinical reasoning. Data are a collection of unorganized or uncontextualized objective facts, while information is contextualized data that convey values, meanings, and purposes. Knowledge, on the other hand, is a mental abstraction of a given situation, and can be viewed as a comprehensive understanding of a given subject that, in the case of clinical practice, enables clinical judgments, decision making, and goal-directed activities [28].

There is a hierarchal bidirectional relationship among data, information, and knowledge, and this comprises a bottom-up and a top-down direction. In the bottom-up hierarchy, cognitive information-processing activities include capturing situational data and gathering basic informative elements (perception stage), processing data into usable information that is then transformed into clinical-knowledge-based rules and patterns that support clinical judgments and decision making (comprehensive or understanding stage), and predictive analysis (projection stage), all of which enable the execution of goal-directed clinical tasks (Figure 2) [28,29].

In the top-down hierarchy, the relevant predetermined knowledge, past experience, putative plans and goals, and established expectations are prerequisites for the selection, analysis, and interpretation of the collected data and for how the information-processing operation is performed [28]. Thus, depending on the given clinical situational actualities and on the clinician’s operative goals, the information-processing operations that bring about SA will be top-down (goal-directed), bottom-up (data-directed), or a combination of both.

Three information-processing cognitive modes, namely, skill-, rule-, and knowledge-based modes, which differ in relation to the degree of the conscious control employed for their operations, are used to develop SA and clinical reasoning (Figure 3). Skill-based activities are highly automated and are performed with minimal conscious awareness; rule-based activities use a higher level of conscious control and are driven by predetermined mental models, standardized subordinates, and stored rules and guidelines that have been developed in association with similar previously managed clinical situations; knowledge-based activities employ the highest level of conscious control, which necessitates the de novo generation of plans and goal-directed decisions and of mental modes on an ad hoc basis through deliberate, time-consuming, analytical, and cognitively effortful processes (Figure 3) [19,30,31].

The use of clinical decision support systems and electronic health records provides clinicians with important patient-related information and with computer-generated clinical knowledge that includes risk assessments, risk reduction factors, clinical-knowledge-based rules and patterns, and predictive analyses of probable post-intervention clinical outcomes. This information and knowledge are generated from the patient’s medical history and through data mining, machine learning techniques, and statistical modeling [29]. Machine learning is a branch of artificial intelligence focusing on developing computer systems that can learn the patterns and relationships in large volumes of data in order to formulate classifications and predictions; statistical models are mathematical representations (mathematical models) that, when applied to data (statistical modeling), can identify and analyze correlations between different variables and draw inferences and predictions from the patterns in the data. The results of these computer-driven data/information/knowledge-processing techniques may ease and enhance the outcomes of clinicians’ own information-processing mechanisms with subsequently improved clinical judgments, decision making, and SA (Figure 4).

## 4. Clinical Judgments and Decision Making in Relation to Situational Awareness

The cognitive processes that are required to develop an accurate clinical SA force clinicians to focus their attention and to allocate resources of mental energy to the given situational elements and then to analyze and interpret the captured data and perceived information in the context of the situational constraints and of the performance of the goal-directed tasks. Goal-directed tasks may be pre-determined, and in such cases, the clinician’s role is to identify situational cues that are necessary for the efficient and effective execution of the prescribed plan (top-down); or, in the case of goal-directed activities that still need to be constructed, the clinician has to recognize situational cues that will enable the categorization and conceptualization of the current situation in accordance with previous similar mental situational models. This pattern recognition process facilitates clinical judgments, decision making, and the planning of appropriate goal-directed activities (bottom-up) [21] (Flach, 1995) [19] (Figure 4). The conscious activity of information processing and the subsequent analysis of, and decision on, the actual significance of the interacting elements of the clinical situation determine the properties of the given situation and the accuracy of the SA [21].

The cognitive pathways involved in information processing in relation to solving clinical problems comprise a non-conscious, automatic, intuitive, and fast reasoning pathway (system 1), as well as a conscious, deliberate, analytical, effortful, and time-consuming reasoning pathway (system 2) [22,32,33] (Figure 4).

In the context of clinical practice, a given clinical problem may be typical, atypical, or complex, and it may display elements of uncertainty and/or ambiguity [16,22,34]. Depending on the nature of the clinical situation and on the cognitive functions required for driving the decision-making process, system 1 and system 2 may either operate sequentially with system 1 being the first in action, followed by the analytical and deliberate system 2, which monitors and, if necessary, corrects the intuitive judgments and decisions constructed by system 1, or they may operate concurrently or interchangeably [16,22,32,34] (Figure 4).

The intuitive reasoning pathway (system 1) that is used to formulate clinical judgments and decisions is based on the experience and expertise of the clinician, on heuristics, and on pre-determined mental models. The mental representations of a given current clinical situation’s elements and circumstances are matched with previous patterns and knowledge of clinical situations stored in the memory by using a pattern recognition process, and each identified match contributes to the understanding of the current clinical situation; the pre-determined mental models are revised and then adapted to accommodate the given current clinical situation. This process augments the clinician’s repertoire of mental models and facilitates fast decision making and the prediction of future situational eventualities [5,7,9,22,34]. Thus, comprehensive domain-specific knowledge and previous clinical experience are essential for the development of accurate SA and for the management of the given clinical situation [5].

However, in the face of deficiencies in prior knowledge and mental models, as is the case with novices and inexperienced clinicians, intuitive reasoning and heuristics may be ineffective in solving clinical problems [11,34]. In such circumstances, or when a clinical problem is compound or atypical, the clinician has to employ the deliberate and analytical reasoning pathway of system 2, which is characterized by its time-consuming and complex information-processing operations that require the use of critical thinking, statistical concepts, deductive logic, scientific methods, and focused attention [33] (Figure 3). In general, experienced clinicians demonstrate better memory recall and depth of information searching than inexperienced clinicians do [35].

Heuristics and intuitive reasoning are susceptible to cognitive biases. Some common cognitive biases occurring in the context of clinical practice include overconfidence, interpretation of new information that confirms prior beliefs and preferred concepts (even if they are unsubstantiated), formulating clinical judgments on the basis of subjective first impressions or on weak clinical and statistical evidence, vulnerability to the way the situational information is presented (framing effect), underestimating the role of chance and uncertainty, the tendency to focus attention on the obvious situational elements but missing less obvious ones, and terminating the clinical reasoning process before it has been ‘fully completed’ [32,33,36,37]. These cognitive biases are important factors that may interfere with effective clinical judgments, decision making, and the development of SA.

## 5. Emotions as They Relate to Situational Awareness

Emotions may either support or hinder cognitive functioning [38,39], and it appears that the emotional state of a clinician may influence whether the gathered situational information will be processed intuitively or analytically [40]. While a positive affect may support information processing, clinical judgments, and decision making, a negative affect may interfere with sound reasoning, with assessments of the risks of clinical practices, and with weighting the relative clinical importance of situational elements, thus distorting the mental image in SA [16,38,40].

Situational uncertainties and complexities, uncertainty intolerance, worries about inflicting harm to patients, concerns about possible malpractice liability claims, worries about managing a patient’s unrealistic expectations, and a lack of confidence in one’s operational competence are some contextual stressors in clinical practice that may bring about negative emotional responses, such as frustration, anger, agitation, and fear. These emotional responses may have repercussions for the ability to focus attention and for executive functioning, which is critical for the development of SA, clinical judgments, decision making, and goal-directed activities [38,41].

Fatigue, financial crisis, poor personal health, and intrinsic or substance-induced negative feelings and moods are some non-contextual stressors that may have a negative impact on SA and clinical practice, and they are not dissimilar to the effects described above with regard to contextual stressors [14,16,38]. Some of these non-contextual negative emotions are incidentally induced, typically persist without conscious awareness (‘carryover incidental emotions’), and may affect subsequent clinical judgments and decision-making processes [38].

Mental energy is an abstract construct with dynamic properties; it can be viewed as an intrapsychic resource that powers psychological mechanisms including cognition, emotions, motivation, willpower, and executive functioning, all of which are essential for performing goal-directed tasks. Since the resources of mental energy are finite and since significant mental energy is necessary to regulate and control negative emotions by one’s self, the remaining energy available for effortful focused attention, working memory, cognitive flexibility, and information processing, which are critical for clinical reasoning and for the development of SA, is diminished. This negative emotion-induced reduction in resources of mental energy may thus compromise effective clinical reasoning, goal-directed performance, and the development of SA [14,42].

There are considerable variations in clinicians’ psychological responses to similar contextual and non-contextual emotion-generating stressors in clinical practice. These are related, in part, to differences in person-specific cognitive appraisal mechanisms and in other executive functions that influence the experience of and response to emotions. These differences may also account for the different SA experienced by different clinicians in relation to the same clinical problem [43].

## 6. Interventions That May Boost the Development of SA

Managing SA errors in clinical practice and reducing their occurrence should not be considered the sole responsibility of clinicians. Both clinician- and organization-directed measures are required to improve clinicians’ ability to develop accurate SA and reduce the risk of SA errors [14]. Healthcare organizations should introduce, and clinicians should use, health information technology that enables easy access to relevant and up-to-date information, expert second opinions, digital images, clinical guidelines, and algorithms [16]. The successful integration of bioinformatics, clinical data, and information into relevant knowledge and the effective use of clinical decision support systems and electronic health records may simplify the weighing of diagnostic probabilities and their primacy, and they may bring about essential knowledge that is directly applicable to patient-specific clinical situations [44]. As these digital tools can also alert clinicians about patient-specific drug interactions, errors in drug dosing, drug allergies, and alarming test results [44,45,46], their routine use may promote clinicians’ SA and patient safety.

To boost SA acquisition and avoid SA errors, clinicians should have adequate domain-specific structural knowledge and clinical expertise, and they should master both intuitive and analytical cognitive operations that, in turn, support information processing and clinical reasoning, as well as the identification of common cognitive biases that have a negative impact on the development of accurate clinical SA [16]. Thus, clinician-directed interventions for promoting the development of SA should aim at improving domain-specific knowledge, clinical reasoning, and clinical competence, at improving communication and time-management skills and the understanding of statistical concepts, and at enhancing psychological coping and resilience capacities. This may be achieved through regular continual education and professional development, frequent engagement in mindfulness and meditation, frequent reappraisals of inaccurate SA in current and past clinical situations, and regular metacognitive and reflective practices.

In the context of SA, meditation, mindfulness, and metacognition may enable clinicians to increase their critical awareness and understanding of the complexities of a given clinical situation and to monitor and evaluate their clinical judgments and rational reasoning so that decision making and goal-directed activities can be modified according to the situational dynamics. Regular metacognitive and reflective practices also have the capacity to identify and minimize the effects of cognitive biases on the accuracy of SA, clinical judgments, and decision making [16].

Interventions driven by healthcare systems aiming at boosting clinicians’ SA, thus reducing the frequency of SA errors include the following. Firstly, mechanisms for supporting reporting and documenting incidences of SA errors should be established. Secondly, whenever possible, clinical work overload, administrative duties, and clerical responsibilities should be reduced. Thirdly, management should recognize and emphasize the importance of acquiring high-level clinical judgment and decision-making skills, should support continuing education and professional development, and should institute structural, managerial, and cultural modifications that promote autonomy, competence, relatedness, comradery, and communication among the members of the clinical team at the workplace [14,16]. Implementing all of these measures has the capacity to facilitate development SA and reduce the incidence of SA errors.

## 7. The Way Forward

Situational awareness and clinical performance are dictated by clinicians’ constraints (e.g., working memory, relevant mental models, information-processing capacity, mental acuity), situational constraints (e.g., complexity, atypicality, uncertainty, lack of information), and the dynamic relationship between the two [23]. However, as SA errors are not uncommon and may jeopardize patient safety, measures have to be introduced to improve the capacity of clinicians to develop high-level SA [26,27]; as there is insufficient evidence-based information about the prevalence and incidence of clinical-practice-related SA errors and about extrinsic and intrinsic factors that influence their occurrence, it is prudent to develop and institute mechanisms that facilitate the recording and accumulation of evidence-based information about the occurrence of SA errors. This may enable the planning and implementation of interventional policies aimed at addressing this potential healthcare problem.

In order to increase the awareness of healthcare providers about the important role that SA plays in everyday clinical practice, basic knowledge regarding this construct should be included in the curricula of undergraduate and postgraduate medical, dental, and nursing education, as well as in continuing education courses and relevant healthcare-related conferences [16]. To the best of our knowledge, in most clinical disciplines, the skills of acquiring SA are primarily imparted to trainees through tacit learning during supervised clinical practice, but not through designated well-structured teaching modules. Establishing formal training platforms using simulation methods featuring clinical cases—both typical and atypical—can increase clinicians’ capacity to develop accurate SA and thus minimize the frequency of clinical-practice-related SA errors [27].

More research is needed to acquire a clear understanding of the neural mechanisms that generate and regulate mental energy and of the role that mental energy plays in cognition, emotion, mood, alertness, and information processing [33,47]; to develop mechanisms that can identify specific emotions that are detrimental to the process of clinical reasoning; and to establish the best approach to minimizing the adverse effects of negative emotions on cognitive processes that drive the development of SA [48].

There is a pressing need to design and develop cognitive engineering systems that enhance the integration of situational information and, thus, facilitate the formulation of coherent mental models that support clinical reasoning, diagnostic accuracy, and the prediction of changes in situational events. Augmented cognition technology may boost decision makers’ abilities to master complex and uncertain clinical situations and to manage them more efficiently and effectively. Such cognitive systems may also counteract the negative impact that data- and information-processing loads have on executive functioning and counteract the effects of communication problems arising between clinicians [2,4,19,26,27,49]. It is also important to determine the influences that different personality traits of clinicians may have on the acquisition of SA and what the best methods for managing clinical uncertainty and improving clinical reasoning are [16]. Subsequently, all of these research topics may facilitate information processing, clinical judgments, decision making, and SA development.

## 8. Conclusions

Situational awareness plays an essential role in the complex and dynamic process of decision making and operational activities in clinical practice. It facilitates the dynamic adaptation of goal-directed plans in response to changing eventualities, and it enables the anticipation of future situational dynamics. Domain-specific experience and expertise and cognitive acuity are essential for developing high-level clinical SA. Situational awareness is susceptible to information overload, cognitive bias, and rapidly changing situational actualities; adequate resources of mental energy are required for its effective development [5]. In order to acquire the skills necessary to develop accurate SA, clinician trainees have to be repeatedly exposed to simulations of complex and dynamic high-fidelity real-life clinical situations.

## Figures and Tables

**Figure 1 healthcare-11-03098-f001:**
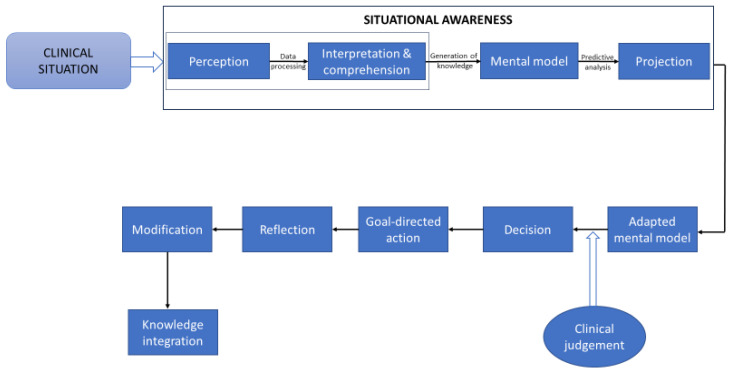
A simplified diagram showing the cognitive information processing pathway that is used in the development of situational awareness, decision making, and clinical operations. The ‘perception’ component of this pathway captures the situational characteristics obtained from the patient’s history, physical examinations, and diagnostic test results. The interpretation and comprehension of the perceived information are determined by using pattern recognition processes, by using the clinician’s cognitive skills, memory capacity, and domain-specific competence, and by considering expert and second opinions. All of these factors are essential for constructing situational mental models, and comprehensive knowledge of the situational constraints enables the formulation of differential and working diagnoses, as well as the performance of adaptive goal-oriented activities [6]. The ‘projection’ component deals with situational predictive analyses and with the search for additional situation-related and relevant scientific information, with the consideration of alternative treatment options, and with forecasting treatment outcomes and ongoing changes in situational eventualities [6]. Based on the acquired situational awareness and adapted ‘mental model’, a decision is made, and it is followed by the execution of the planned goal-directed tasks. Subsequently, the process of reflection in relation to the operational plans and tasks is conducted. Finally, the newly acquired knowledge is integrated within pre-determined mental models [7,9,15,22].

**Figure 2 healthcare-11-03098-f002:**
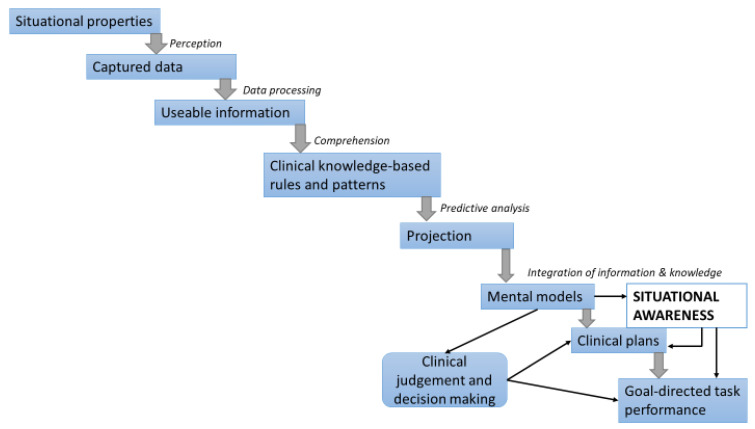
Bottom-up information processing pathway that enables clinical judgments, decision making, choices, and adaptive goal-directed clinical operation [28,29].

**Figure 3 healthcare-11-03098-f003:**
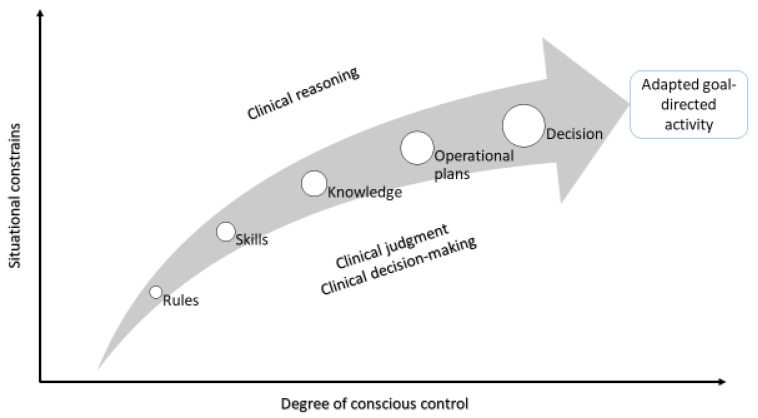
A simplified diagram showing the relation between the cognitive depth employed and the constraints (uncertainty, complexity, deficient information, etc.). Adapted from Cumming, 2018 [30].

**Figure 4 healthcare-11-03098-f004:**
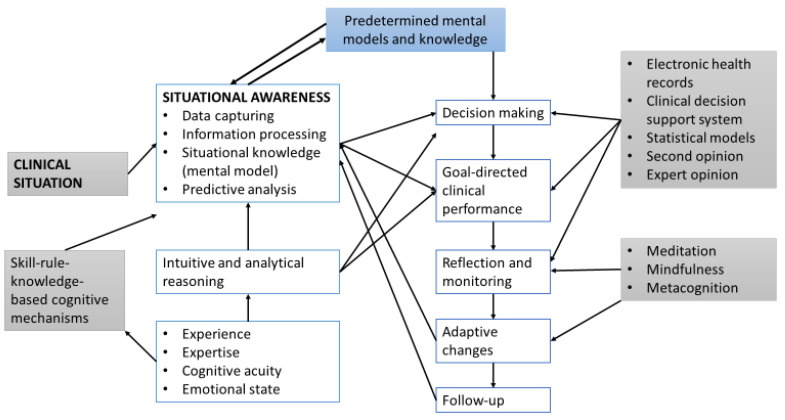
A simplified flowchart illustrating some of the cognitive processes employed in the development of SA, in clinical decision making, and in goal-directed activities [7,22].

## Data Availability

Not applicable.

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
