# Peer review of "Situational Awareness in the Context of Clinical Practice"

_healthcare, 2023, doi:10.3390/healthcare11233098_

Round 1

Reviewer 1 Report

Comments and Suggestions for Authors

The authors aim to review ‘Situational Awareness in the Context of Clinical Practice’.

This work may have value for some clinical readership and sections of the healthcare administrators and policy makers. Although, the reviewer recommend significant restructuring and editing of this manuscript.

Introduction section requires direction. 

The introduction sections in paragraph 1, 2, 3, 4, 5 in current form are complex, long and winding, and repetitive at times.

The authors should start by introducing the readers to a simplified definition of situational awareness, background, history, the key relevant clinical areas it is valuable in and its goals/purpose. What are the guidelines or commentary by major health bodies in this area, e.g., WHO?

Paragraph 6 talks of factors leading to issues with robust situational awareness but also ends with a potential solution though utilization of electronic health records etc., before reconverging to issues in paragraph 7 and 8. Please maintain consistency is first presenting the factors affecting situational awareness, followed by any potential solution.

In the final section, please briefly introduce the readership to current gaps in knowledge.

The authors mention – ‘The purpose of this narrative review is to contribute to a better understanding of the role that SA plays in  the context of clinical activities, including diagnosis, planning, decision-making and execution of goal-directed clinical operations’, although in subsequent sections the authors discuss factors associated with situational awareness. This is confusing, please reframe the aims of this review appropriately.

Recommend simplifying the text and restructuring the manuscript linearly. See example as below.  The authors may use this or a similar format. 

1.     Background/Introduction – Briefly focus on simple definition, history, relevant background, milestones and gaps.

2.     SA (i. Construct, ii. Procedures)

3.     Factors affecting SA (i. Internal, ii. External factors).

4.     Factors effected by SA (Clinical Decision Making, Diagnosis, Treatment planning etc).

5.     Interventions/Factors improving SA.

6.     Future Directions.

Other changes –

1.     Please maintain consistency in abbreviations (including abstract), e.g., situational awareness is mentioned either as SA or unabbreviated interchangeably, throughout text.

2.     Text accompanying Figure 2, 3, 4 should be place as a cation after figure (as with figure 1), not within figure.

3.     There are 50 references in total, and 16% self-citations. No sure why #43 is cited for the preceding text. Some other references can be replaced with more pertinent citations as well.

Comments on the Quality of English Language

While the quality of English is acceptable, the manuscript was a laborious read. Multiple sentences were overtly long. Recommend appropriate simplification of the text and language.

Author Response

Dear Editor,

Thank you and the reviewers for the hard work on reviewing our manuscript.

RESPONSE TO REVIEWER 1

Other changes: 

  • As requested, we consistently throughout the text refer to situation awareness as SA.
  • We have placed all the legends of the Figures as is with Figure 1.
  • Reference number 43 was introduced because in their article, Feller et al., deal with how emotional states affect mental energy and subsequently cognitive function, and this has relevance to the development of accurate SA.
  • All the self-citation references deal with elements of SA, such as mental energy, clinical judgement, clinical decision-making, uncertainty, etc.
  • With regard to the other comments, it seems that the reviewer would like us to turn our article ‘on its head’
    • We applied a great deal of thought and hard work in the process of constructing and in the write-up of this article, and we are of the opinion that changing the format and references will weaken rather than strengthen the article.
    • We are also of the opinion that the text under the current subheadings follow a logical order, and the information as presented, will promote the subject matter knowledge of the reader (the healthcare practitioner).
    • We apologise about our style of writing that comprise of long sentences and are at times quit involved.
    • Indeed, SA is a complex construct that is difficult to express in a simple way; and when one tries to simply a complex concept, one runs the risk of generating an oversimplification that may distort the actual meaning of the concept (SA), thus misleading the reader.
    • Although reading the text of this article may be demanding or ‘heavy’, we believe that it is informative and useful.

Thank you very much for your hard work reviewing this article.

Yours Sincerely

Razia Khammissa

Reviewer 2 Report

Comments and Suggestions for Authors

An innovative perspective (SA) but one that requires additional clarification. There is the tendency to discuss the concept in analytical terms without understanding the uniqueness of each patient. In any patient situation, there is always a flow of information and thinking, you appear to assume that it is uni-directional. I hope that my provided comments are useful.

Author Response

Thank you and the reviewers for the hard work on reviewing our manuscript.

 RESPONSE TO REVIEWER 2

  1. As requested we have elaborated on the ‘boundaries’ of this review and on the method by which the data and information were obtained. Please see page 3, last paragraph under the heading ‘Introduction’ (highlighted in yellow) that reads:

‘The information for this narrative review was obtained by employing PubMed and MEDLINE database search engines using the search terms situational awareness, information processing, pattern recognition, mental models, clinical reasoning, clinical judgement, clinical decision-making, medical uncertainties, analytical reasoning, intuitive reasoning and health information technology; and by analyzing references from relevant English language articles that were deemed pertinent.’

  1. As requested, we have also clarified the concept of ‘it also enables projecting….’

According to the dictionary, ‘projection’ can be defined as an estimate or forecast of a future situation based on a study of present trends (google dictionary) or as a calculation or guess about the future, based on the current information that one has (online Cambridge dictionary). So with these definitions that are common knowledge, we have modified the text on lines 57-59 to read:

‘[6,11,12]; and it enables the anticipation of evolving changes in the current situational actualities and task description, based on similarities to past clinical situations. It also enables projecting the risk-benefit-trade offs of the current operational activity to future clinical situations [5].’

We hope that this change will help in clarifying the matter.

Furthermore, the term ‘projection’ is also dealt with in line 122 that reads: ‘…….[6]. The ‘projection’ component deals with situational predictive analysis and with search of additional situation-related and relevant scientific information;…..’

3.1          We have dealt with the term projection in point 2 above.

3.2          ‘Mental model’ can be viewed as an internal representation of knowledge about an external situation. We have added this definition to the text. Please see lines 94-95 that now reads:

‘………..to the development of mental models (a mental model is internal representations of and knowledge about an external situation and to planning and executing goal-directed tasks) [19].’

3.3          Perception according to the online Cambridge dictionary, is a belief or opinion formed by using one’s senses (this should be common knowledge for any healthcare professional acquired in early years of academic studies). Thus situation-specific perception may differ among individuals, between clinicians and their patients, between different clinicians and between different patients. Therefore as the reviewer states the ‘assessment’ of a patient may change with time and treatment outcome. With this definition in mind, perception involves senses, and cognitive/emotional activity that drives clinical judgement and decision-making occur subsequently to perception.

3.4          The reviewer asks: ‘…… is situational awareness a process or an outcome?’

The answer is: it is an outcome. It is explained in the text on lines 130-137 that reads:

‘Thus, SA should be regarded only as a description or label of an internal representation of any given situation, and not in itself a cognitive mechanism or process [9, 21, 22]. The cognitive mechanisms that play active roles in developing a meaningful conscious awareness about a given situation include long-term memory, short term memory, attention, and other executive functions, but not SA itself [7]. Situational awareness is not an empirical reality but an abstract construct; and a failure of a clinical operation associated with deficient SA cannot be attributed to the ‘impaired’ SA but rather to the imperfect functional activity of the psychological mechanism that generated it [23].’

4             Page 6, line 211 ‘machine learning technology’ and ‘statistical modeling’. As requested, we have defined these terms. Please see page 6, line 212 that reads:

‘Machine learning is a branch of artificial intelligence focusing on developing computer systems that can learn the pattern and relationships in large volume of data in order to formulate classifications and predictions; and statistical models are mathematical representations (mathematical models) that when applied to data (statistical modeling) they can identify and analyse correlations between different variables, and draw inferences and predictions from the pattern in the data.’

5             Page 6: ‘I would suggest that knowledge comes before skills in the Figure.

We respectfully disagree, please see references 19, 29 and 30 and the explanation given in lines 194-204 that reads:

‘Three information-processing cognitive modes, namely skill-, rule- and knowledge-194 based, that differ in relation to degree of the conscious control employed for their operations, are used for developing SA and clinical reasoning (Figure 3). The skill-based activity is highly automated and is performed with minimal conscious awareness; the rule-based activity uses a higher level of conscious control, and is driven by predetermined mental models, standardized subordinates, stored rules and guidelines which have been developed in association with previously managed similar clinical situations; and the knowledge-based activity employs the highest level of conscious control that necessitates the de-novo generation of planes and goal-directed decisions, and of mental modes on adhoc basis, through deliberate, time-consuming, analytical, cognitively effortful processes (Figure 3) [19, 29, 30].

  1. Page 7, Figure 4 ‘all the arrows are unidirectional….’ Thank you! It was an error on our side. Please see the corrected Figure.

  1. Page 9, 10 and 11: under the headings ‘Interventions that may boost the development of SA’ and ‘The way forward’, we discuss: clinician-organisational-educational measures that are required to improve the clinician’s ability to develop accurate SA. The authors are of the opinion that all these three elements under the umbrella of ‘Healthcare structure’ are equally important in promoting SA.

8             ‘I would strongly disagree…………..’

While effective simulation programs are indispensable, tacit learning is also important. We deleted the word ‘establishing’ on line 378, and introduced the word ‘employing’

9             I enjoyed reading the manuscript……very analytical in tone and structure.’

  • We are pleased that you find the article interesting.
  • Indeed, the article is quite abstract, and analytical in tone and structure, but the topic is very complex, and trying to simplify the discussion, carries a risk of oversimplification which may distort the actual meaning of SA and subsequently lead to misunderstanding.

10           Thank you!

Yours Sincerely

Razia Khammissa

Round 2

Reviewer 1 Report

Comments and Suggestions for Authors

Thank you for your responses to the review and making the minor edits.

Comments on the Quality of English Language

Minor editing of language and copy editing is required.

Author Response

Thank you for your hard work reviewing the manuscript.

Reviewer 2 Report

Comments and Suggestions for Authors

I appreciate the opportunity to review this revised manuscript. There remains a stilted writing context to this manuscript.

Comments on the Quality of English Language

While the authors do appear to be able to write in English - the style is stilted, and this will certainly influence the reader's enjoyment and learning. 

Author Response

Thank you for your hard work reviewing our manuscript.
